# Adaptive Mechanisms of Renal Bile Acid Transporters in a Rat Model of Carbon Tetrachloride-Induced Liver Cirrhosis

**DOI:** 10.3390/jcm11030636

**Published:** 2022-01-27

**Authors:** Chiara Donadei, Andrea Angeletti, Maria Cappuccilli, Massimiliano Conti, Diletta Conte, Fulvia Zappulo, Alessio De Giovanni, Deborah Malvi, Rita Aldini, Aldo Roda, Gaetano La Manna

**Affiliations:** 1Nephrology, Dialysis and Renal Transplant Unit, IRCCS—Azienda Ospedaliero-Universitaria di Bologna, Alma Mater Studiorum University of Bologna, 40138 Bologna, Italy; chiara.donadei@studio.unibo.it (C.D.); andrea.angeletti@studio.unibo.it (A.A.); maria.cappuccilli@unibo.it (M.C.); diletta.conte2@unibo.it (D.C.); fulvia.zappulo@studio.unibo.it (F.Z.); 2Division of Nephrology, Dialysis and Transplantation, IRCCS Istituto Giannina Gaslini, Genoa Largo Gaslini, 16148 Genoa, Italy; 3Department of Pharmacy and Biotechnology, Alma Mater Studiorum University of Bologna, 40138 Bologna, Italy; massimiliano.conti8@studio.unibo.it; 4Department of Experimental, Diagnostic and Specialty Medicine—DIMES, “F. Addarii” Institute of Oncology and Transplant Pathology, Alma Mater Studiorum University of Bologna, 40138 Bologna, Italy; alessiodegiovanni@gmail.com (A.D.G.); deborah.malvi@aosp.bo.it (D.M.); 5Department of Chemistry “G. Ciamician”, Alma Mater Studiorum University of Bologna, 40126 Bologna, Italy; rita.aldini@unibo.it (R.A.); aldo.roda@unibo.it (A.R.)

**Keywords:** acute kidney injury, bile acids, choleric nephropathy, liver cirrhosis, organic transport proteins, rat model, serum inflammation biomarkers, urinary tubular injury biomarkers

## Abstract

Background: Acute kidney injury (AKI) is common in advanced liver cirrhosis, a consequence of reduced kidney perfusion due to splanchnic arterial vasodilation and intrarenal vasoconstriction. It clinically manifests as hepatorenal syndrome type 1, type 2, or as acute tubular necrosis. Beyond hemodynamic factors, an additional mechanism may be hypothesized to explain the renal dysfunction during liver cirrhosis. Recent evidence suggest that such mechanisms may be closely related to obstructive jaundice. Methods: Given the not completely elucidated role of bile acids in kidney tissue damage, this study developed a rat model of AKI with liver cirrhosis induction by carbon tetrachloride (CCl_4_) inhalation for 12 weeks. Histological analyses of renal and liver biopsies were performed at sacrifice. Organic anion tubular transporter distribution and apoptosis in kidney cells were analyzed by immunohistochemistry. Circulating and urinary markers of inflammation and tubular injury were assayed in 21 treated rats over time (1, 2, 4, 8, and 12 weeks of CCl_4_ administration) and 5 controls. Results: No renal histopathological alterations were found at sacrifice. Comparing treated rats with controls, organic anion transporters were differentially expressed and localized. High serum bile acid values were detected in cirrhotic animals, while caspase-3 staining was negative in both groups. Increased levels of serum inflammatory and urinary tubular injury biomarkers were observed during cirrhosis progression, with a peak after 4 and 8 weeks of treatment. Conclusions: These findings suggest possible adaptive tubular mechanisms for bile acid transporters in response to cirrhosis-induced AKI.

## 1. Introduction

Physiological changes occurring in patients affected by decompensated liver cirrhosis with ascites leads to a markedly increased risk of acute kidney injury (AKI), with prevalence ranging from 14% to 50% [1,2]. The most frequent causes of cirrhosis-induced AKI include pre-renal injury, acute tubular necrosis (ATN), hepatorenal syndrome (HRS), and post-renal obstruction, the latter representing less than 1% of cases [3]. The widely accepted hypothesis for the pathogenesis of AKI in cirrhotic patients is that renal dysfunction results from peripheral arterial vasodilation with subsequent hemodynamic impairment and intrarenal vasoconstriction [4]. In spite of some experimental findings to support such a pathogenetic link, other contributing mechanisms might be feasibly implicated [5,6]. Of note, patients with similar stage of liver disease and hemodynamic failure can develop AKI or not, suggesting the need of further investigations to explain the occurrence of AKI. Several evidence suggest a possible role for bile acids in inducing choleric nephropathy, a comprehensive term referred to renal dysfunction in the course of obstructive jaundice. Previous reports have described cholestatic liver diseases associated with progressive tubulointerstitial nephropathy that share some features, including the early onset of tubulointerstitial nephritis during the development of cholestasis. More in detail, the increase in serum bile acids levels and renal injury correspond to histological findings of tubular atrophy and dilatation, as well as interstitial and periglomerular fibrosis [7,8,9]. Bile acids are mainly excreted with urine, limiting the toxic serum accumulation [10]. Due to the high hydrophobicity, bile acids promote injury on the lumen surface of tubular cells, contributing to worsening AKI during liver failure [11]. Inflammatory syndrome, typical of liver cirrhosis, may result as an additional mechanism of AKI [12]. A full comprehension of AKI in cirrhosis is limited by the lack of suitable experimental models. To the best of our knowledge, renal injury has been described in rodent models of common bile duct ligation (CBDL) [13] or hepatic ischemia-reperfusion liver injury [14]. Additional animal models of renal damage associated with liver failure include carbon tetrachloride (CCl_4_) abdominal injection in rats; however, it is characterized by multi-organs (brain, bowel, kidney, heart) fibrosis usually occurring after CCl_4_ injection [15,16,17]. In contrast, thioacetamide has been extensively used to generate in animal models of acute, non-chronic liver injury such as cirrhosis [18].

Here we developed a rat model of decompensated liver cirrhosis induced by CCl_4_ inhalation for 13 weeks. The main goal of our investigation was to characterize the functional and histological renal phenotype and to describe the localization of bile acids and organic anions transporters in the kidney, as well as the modulation of circulating inflammatory cytokines and urinary biomarkers of tubular kidney injury during the onset of AKI in liver cirrhosis.

## 2. Materials and Methods

### 2.1. Animals and CCl_4_-Induced Liver Cirrhosis

All the experiments were performed in accordance with the guidelines of the EEC Directive 86/609 [19]. The protocol was approved by the Ethics Committee of the University of Bologna (Protocol code: CES 25.57/80). Thirty-three male Wistar Han rats (Charles River Laboratories, Calco, Italy), weighting 225–250 g, were housed for 3 weeks prior to the beginning of the study, 28 of them for the cirrhosis model and 5 controls. The conditions were kept stable before and during the study: controlled temperature (22–24 °C), standard 12 h light/dark cycle (lights on at 07.00 a.m.), and free access to food and water. The rats of the cirrhosis group were administered with phenobarbital (0.3 g/L in drinking water) for 1 week, and then CCl_4_ was provided through inhalation over 12 weeks [20]. Phenobarbital was continued throughout the study as a monooxygenase and conjugating enzymes inducer [21,22]. The rats were placed in a gas chamber (70 × 25 × 30 cm) and compressed air, bubbling through a flask containing CCl_4_, was passed into the gas chamber via a flow meter (1 L/min). The animals were exposed to the gas atmosphere twice a week, starting with 0.5 min of bubbling and 0.5 min in the gas atmosphere. Afterwards, the time was increased to 1 min and then by 1 min until 5 min of air flow and 5 min in gas atmosphere were reached. Seven rats of the cirrhosis group died during the 7–12-week period of CCl_4_ treatment.

Serum samples were weekly stored and at the sacrifice. Moreover, the animals were weighted and monitored according to the humane endpoints for research animals set by the Ethical Committee of the University of Bologna, namely: body weight changes, external physical appearance, behavioral changes and physiological changes (e.g., body temperature, hormonal fluctuations, and clinical pathology).

### 2.2. Histology

Renal and liver tissues for histological analysis were collected at sacrifice. Renal biopsies were evaluated blindly and independently by two pathologists and a nephrologist, experienced with laboratory rodents. Slices were fixed in 4% formaldehyde for histopathological analysis. Paraffin sections (1 µm thick) were stained with hematoxylin and eosin, trichrome and PAS. Liver histology was evaluated in blinded fashion by two pathologists. Slices were fixed in 4% formaldehyde and paraffin sections (1 µm thick) were then stained with hematoxylin and eosin.

### 2.3. Immunohistochemistry

The following bile acids and organic anion tubular transporters were studied by immunohistochemistry: NTCP (solute carrier family 10 member), bile salt export pump (BESP), P-Glycoprotein (P-Gly), ASBT (solute carrier family 10 member 2), multidrug resistance protein 4 (MRP4), osteopontin, caspase-3, and Toll-Like Receptor 4 (TLR4).

Briefly, all the bioptic specimens were fixed in formalin and embedded in paraffin. From paraffin blocks 3-µm-thick sections were cut, deparaffinated with Bio Clear (Bio-Optica, Milan, Italy) and hydrated with a decreasing alcohol sequence (100%, 96%, and 70%) plus distilled water. The block of the endogenous peroxidases was obtained using a bath in 3% H_2_O_2_ in a methanol solution for 10 min. The antigen retrieval was carried out through a citrate buffer at pH 6.0, 4 cycles of 5 min each in 750 W microwave. The slides were then equilibrated in the Tris Buffered Salin buffer (1X TBS pH 7.6) for 10 min, before incubation with the primary antibodies. The main technical features of the antibodies used for immunohistochemistry are detailed in the Appendix A.

Each slide was incubated with the primary antibodies in a wet chamber at room temperature for 1 h, washed three times (5 min each) with the TBS buffer, and finally incubated with Novolink Polymer Detection Systems (Leica Biosystems Newcastle Ltd., Newcastle upon Tyne, UK), following the manufacturer’s instructions. Finally, the slides were counterstained with Hematoxylin (2 min), dehydrated with an increasing alcohol sequence (70%, 96%, and 100%), and mounted.

For the purposes of the study, the immunoreactivity in the proximal tubular kidney cells was analyzed both qualitatively and semiquantitatively. A semiquantitative score was performed using a four-point scale based on immunohistochemistry staining intensity: +, weak; ++, moderate; +++, intensive; ++++, highly intensive. Some of the images were analyzed quantitatively using the ImageJ software (NIH) and used as reference to allow semiquantitative scoring, as previously described [23].

### 2.4. Kidney and Liver Function Indexes

Blood samples were collected stored at −80 °C at 1, 2, 4, 8, and 12 weeks of treatment with CCl_4_ and at the same time points in controls.

Blood nitrogen (BUN), albumin (ALB), alanine transaminase (ALT), aspartate transaminase (AST), alkaline phosphatase (ALKP), gamma-glutamyl transferase (GGT), total bilirubin (TBIL), ammonia (NH_3_), serum creatinine (sCreat), sodium (Na^+^), and lactate dehydrogenase (LDH) were assayed using commercial kits validated for small animal diagnostics according to the manufacturer’s indications (Catalyst Dx^®^ Chemistry Analyzer, Westbrook, ME, USA).

The qualitative and quantitative analysis of bile acids in serum and urines was performed through a validated HPLC-ES-MS/MS method. The limit of quantification (LOQ) for bile acids was lower than 100 nM.

### 2.5. Luminex Analysis of Cytokines, Growth Factors and Urinary Markers

Luminex technology was applied for a multiplex assay of biomarkers involved in inflammation and kidney injury in urine and plasma through the Luminex MAGPIX^®^ system (Millipore Corp., Billerica, MA, USA). A commercial kit was used for the simultaneous quantification of the circulating plasma levels of IL-1α, IL-1β, IL-2, IL-4, IL-5, IL-6, IL-7, IL-10, IL-12p70, IL-13, IL-17A, IL-18, G-CSF, GM-CSF, GRO/KC, M-CSF, MCP-1, MIP-3α, TNF-α, IFN-γ, RANTES, VEGF, and EPO (Bio-Rad, Bio-Plex Pro™ Rat Cytokine 24-plex Assay #171K1001M). Two kits were utilized to assess nephrotoxicity in urine specimens: one for MCP-1, clusterin, KIM-1, IL-18, and Osteopontin (Kit Rat Kidney Toxicity 1 Panel #171KTR1CK) and the second for B2M, Cystatin C, NGAL and Calbindin (Kit Rat Kidney Toxicity 2 Panel #171KTR2CK).

Standard solutions were prepared for the calibration curve and quality controls according to manufacturer’s protocol. Briefly, seven standards obtained by serial dilutions, quality controls, samples (diluted 1:2), and magnetic beads were loaded into a 96-well plate and incubated overnight to allow binding between the antibody and the molecule of interest. The plate was then rinsed three times and then 50 μL of a cocktail of specific biotinylated antibodies were added to each well and incubated for 1 h. Next, the plate was rinsed again three times to remove unbound biotinylated antibodies, and then streptavidin-phycoerythrin conjugate (streptavidin-PE) was added to each well. Afterward, another wash was performed to remove unbound streptavidin-PE. Beads were resuspended in 100 µL of Sheath Fluid solution. Finally, the plates were read through the MAGPIX Luminex^®^ system (Luminex, Austin, TX, USA) and data were analyzed using the MILLIPLEXR Analyst 5.1.Software (Merck KGaA, Darmstadt, Germany).

### 2.6. Overall Score of Decompensated Cirrhosis

The overall cirrhosis score was defined using histological and fibrosis grading, blood biomarkers, ascites, and encephalopathy.

The liver necroinflammatory score based on the scheme previously developed by Ishak et al. [24] is described in the Appendix A. The fibrosis score, based on staging and architectural changes, is illustrated in the Appendix A according to both Ishak’s scheme and Meta-analysis of Histological Data in Viral Hepatitis (METAVIR) system [25]. The grade indicates the activity or degree of inflammation and the stage represents the amount of fibrosis or scarring.

The biochemical score was based on albumin, total bilirubin, ammonia, and creatinine levels. The biochemical grading was defined as follows: albumin (g/dL) ranged from 0 (3.2 ± 0.2) to 3 (<1.8); total bilirubin (mg/dL) score ranged from 0 (<0.5) to 4 (>3.0); NH_3_ (µmol/L) ranged from 0 to 4 (>300); sCreat (mg/dL) score ranged from 0 (0.4 ± 0.2) to 3 (>2.5). According to this scheme, a total biochemical score was established in a range from 0 to 14.

The ascites score was established at sacrifice. Laparotomy was performed and four strips of absorbing paper (5 × 30 mm) were placed in the abdominal cavity and removed after 3 min. The amount of ascites was calculated as the difference in the strip weight before and after their placement in the abdominal cavity.

The encephalopathy score includes the behavioral changes observed at weekly time intervals at the end of the study. Spontaneous behavior ranged from 0 to 1.5. Evoked responses ranged from 0 to 1.5, for a total of 3 scores.

### 2.7. Statistical Analysis

Data are presented as mean ± standard deviation (SD) for continuous variables and as absolute numbers for categorical variables. Continuous variables were compared by Student *t*-test or Mann–Whitney, as appropriate, and categorical variables were compared using the chi-square test for contingency tables. The possible impact of inflammatory cytokines and urinary biomarkers on renal function (estimated though serum creatinine) was analyzed through Spearman’s rank correlation coefficients. With increasing values of variables (inflammatory cytokines, growth factors, and biomarkers of renal injury), creatinine increases (positive values) or decreases (negative value). A *p* value < 0.05 was considered statistically significant. Statistical analyses were performed using the statistical package Stata (version 14.2, Stata Corporation).

## 3. Results

### 3.1. Kidney and Liver Histopathological Alterations

No major histopathological findings were evident in the kidneys of the CCl_4_-treated rats. Some cases showed mild pathological lesions of tubulointerstitial damage: at PAS stain, mild vacuolization in proximal tubular cells were present in 5 cases (23.8%). With trichrome stain, ID4 and ID9 showed some proximal tubular cells containing lipid droplets and vacuoles (foam cell), with mild structural degeneration and nest type reorganization. The hematoxylin and eosin stain showed mild-to-moderate necrosis areas, in the cortical and medullary compartment in ID4.

As expected, the kidneys from the five control rats did not show pathological signs.

Liver biopsies revealed various degrees of liver decompensation in all the 21 CCl_4_-treated rats, while the 5 untreated controls did not show signs of hepatitis or fibrosis.

After 13 weeks of CCl_4_ inhalation, the livers of the treated animals were consistent with cirrhosis. Livers were of different size and weight (from 5.6 to 16.7 g). Enlarged spleens were always observed (0.9 ± 0.4 g in cirrhosis vs. 0.5 ± 0.5 g in controls).

Table 1 reports the stages of liver activity and fibrosis based on Ishak’s [23] and METAVIR scoring systems [24] in the 26 experimental animals. All the 21 CCl_4_-treated rats showed a cirrhotic (METAVIR F4; N = 18, 85.7) or a pre-cirrhotic (METAVIR F3; N = 3, 14.3) stage of fibrosis.

### 3.2. Immunohistochemistry Findings

Qualitative analysis revealed that all the antibodies had a specific topographic distribution in the normal proximal tubular kidney cells.

As Figure 1 shows, a semiquantitative assessment revealed: (i) weak granular cytoplasmic positivity, more intensive in cirrhosis for NTCP; (ii) weak cytoplasmic positivity with a granular distribution in cirrhotic rats and moderate in controls for BSEP; (iii) continuous and thin immunoreactivity for P-Gly, always along apical membrane, more intensive in controls; (iv) weak positivity for ASBT in both groups; (v) loss of intensity in treated rats and a more intensive positivity in controls for MRP4; (vi) diffuse apical membrane positivity, more intensive in the treated animals for osteopontin.

### 3.3. Biomarkers of Kidney and Liver Injury

At sacrifice, the CCl_4_ treatment group displayed an altered pattern of the biochemical indexes of cirrhosis, namely high cholestasis and low ammonia values, altered liver function and low levels of serum albumin. TBIL and sCreat were increased in the last weeks of treatment. The median percentage increase from T1 to T12 was 112% for sCreat and 167% for TBIL. The details on sCreat and TBIL levels for each rat in the treatment arm are provided in the Appendix A, respectively. Free endogenous, taurine conjugated and total bile acids, measured in serum and urine samples at T12, were significantly higher in treated rats than in controls (Table 2).

### 3.4. Systemic Inflammation and Early Kidney Injury

In Figure 2, serum levels of circulating proinflammatory cytokines and growth factors are reported as percentage variations compared with baseline (considered as 100%). IL-2, GRO/KC and IL-7 presented a steady increase, with a zenith peak at T12 (T12 vs. T0, *p* = 0.030, *p* = 0.015, *p* = n.s., respectively). MCP-1 had a peak value at T8 (T8 vs. T0, *p* = 0.01). IL-1α, IL-1β, IL-4, IL-6, IL-10, IL-12p70, and IL-13 displayed a similar course with a zenith at T4 (T4 vs. T0, *p* = 0.037, *p* = 0.05, *p* = 0.048, *p* = 0.002, *p* = 0.004, *p* = 0.037, *p* = 0.008, respectively).

EPO was characterized by a particular course, different from all the other biomarkers, with a peak at T2 and a decrease at T12 (only significant for T2 vs. T0, *p* = 0.04).

IL-5, IL-18, and VEGF had increased levels between T4 and T8 (T4 vs. T0, *p* = 0.001, *p* = 0.001, *p* = n.s, respectively; T8 vs. T0, *p* = 0.02, *p* = 0.034, *p* = n.s., respectively).

M-CSF, G-CSF and MIP-3α showed a moderate peak at T4 (T4 vs. T0, *p* = 0.013, *p* = 0.012, *p* = n.s., respectively). IL-17A and RANTES levels remained fairly constant during CCl_4_ treatment, and then decreased at T12 (only significant for IL-17A: T12 vs. T0, *p* = 0.001).

TNF-α revealed a very high percentage variation at T2, T4, and T12 (T2 vs. T0, *p* = 0.03; T4 vs. T0, *p* = n.s.; T12 vs. T0, *p* = 0.001). IFN-γ and GM-CSF were undetectable.

As shown in Figure 3, urinary MCP-1, cystatin-C, and osteopontin showed two peaks at T1 and at T8, while clusterin peaks were at T2 and T8. KIM-1, IL-18 and calbindin displayed a significant rise at T1, followed by a gradual return to baseline levels at T8 and then a slight increase again at T12. B2M and NGAL presented a progressive increase reaching the maximum value at T4 and T8, respectively.

Figure 4 is a schematic representation of immunohistochemistry findings. The topography of transport proteins for BA is shown in renal proximal tubule cell of cirrhotic rats and controls, highlighting in particular the different cellular distribution and signal intensity between the two conditions (treated vs. untreated).

As detailed in Table 3, localization and immunoreactivity signals of such transporters were modified in proximal tubule kidney cells in cirrhotic animals compared with controls, while no differences between the groups were reported for Caspase-3 and TLR-4.

### 3.5. Overall Score

The overall score in decompensated cirrhotic rats (ranging from 0 to 30) was obtained as the sum of the scores of histological parameters, biochemical data, ascites, and encephalopathy at sacrifice. Individual values for each rat in cirrhosis group with the related mean ± SD are shown in Table 4. For control rats, the score was 0. The peritoneal fluid was present in all the animals ranging from 1.5 mL to >4 mL; it was blood and bacterial cells free.

### 3.6. Correlations of Serum Creatinine with Blood and Urinary Biomarkers

Table 5 illustrates Spearman’s rank correlation coefficient evaluating the associations of serum creatinine levels, at the different time points, with inflammatory cytokines, and growth factors. A significant positive correlation was found for IL-1α and MCP-1 with sCreat values detected at T2 and a positive correlation for IL-18, IL-6, GRO/KC, and IL-12p70 with sCreat detected at T4 (nearly significant for IL-18).

The associations of sCreat levels at different weeks of treatment with urinary biomarkers of kidney injury are shown in Table 6. Spearman’s rank correlation test revealed a significant positive correlation for IL-18 and osteopontin with sCreat detected at T2 and a positive correlation for B2M and NGAL with sCreat detected at T4.

## 4. Discussion

AKI occurs frequently during end-stage liver disease and in advanced cirrhosis regardless of the triggering causes [26]. The hypothesis of a pathogenetic link between peripheral arterial vasodilation and cirrhosis with the related complications was supported by several hemodynamic studies in humans and experimental models. However, the limitations of both hemodynamic and non-hemodynamic factors to explicate the hepatorenal syndrome development through the peripheral arterial vasodilation hypothesis (PAVH) [6] opens the way to novel challenging clinical scenarios [27].

In the present study, a different approach to advanced cirrhosis was followed, considering the whole spectrum of disease manifestations [12]. PAVH does not fully account for the mechanisms underlying the progressive decompensation steps in cirrhosis, nor recognizes other factors impairing the effective blood volume. Bernardi et al. analyzed in detail the critical points about PAVH, reinforcing the concept that it alone cannot explain all the pathophysiological and clinics events occurring during cirrhosis [12,28,29]. In view of these limitations, novel theories have been proposed, highlighting the needing of new disease models able to represent all the chronic manifestations observed during cirrhosis development. We therefore focused our attention on choleric nephropathy, in an experimental rat model in which CCl_4_ inhalation induced decompensated cirrhosis [30]. Interestingly, we detected a decline in AKI biomarkers levels in the latest treatment weeks that was found to be concomitant with renal bile acids transporter changes. At the end of the treatment, immunohistochemistry revealed a decrease in BSEP and P-Gly and an increase in osteopontin in urinary lumen, a decrease in MRP4 in the basolateral membrane, while in cytoplasmic space NTCP had higher intensity compared with a normal condition observed in untreated control rats. These results are suggestive for tubular transporters adaptive mechanisms, through which kidney increases bile acids excretion during the early phases of liver cirrhosis.

Besides previous data obtained in animal models [31,32,33,34,35,36], there is currently limited clinical evidence for choleric nephropathy in daily practice, particularly in short-term obstructive jaundice and consequent acute tubular necrosis [37]. In jaundiced patients with cirrhosis, van Slambrouck et al. described bile casts in distal nephron segments and hypothesized that the formation of bile casts contributes to kidney injury by direct toxicity of bilirubin and bile itself, as well as by tubular obstruction, with a pathogenetic mechanism similar to that of myeloma or myoglobin cast nephropathy [38]. Therefore, choleric nephropathy emerges as an important factor that may contribute to AKI and/or renal dysfunction, also in patients with chronic liver disease such as cirrhosis [37,38,39].

In order to investigate the possible pathogenic role of bile acids, we performed immunohistochemistry analyses to detect the expression and distribution of bile acids transporters in tubular cells. Current knowledge on the impact of these transporters in kidney function is poor both in healthy and pathologic conditions. Enterohepatic circulation of bile acids is regulated by transport proteins in hepatocytes, cholangiocytes, ileocytes, and renal proximal tubule cells [40,41,42,43,44].

In liver, conjugated bile acids are secreted into the bile by canalicular BSEP, while sulphated or unusual bile acids, high in cholestasis, are carried by MRP2 and P-Gly. Bile acids are stored in the gallbladder and released into the intestinal lumen after meals. Bile acids are then efficiently taken up by ASBT in the ileocytes of the terminal ileum, shuttled by the ileal lipid binding protein in the cytosol to the basolateral membrane of ileocytes and exported by OSTα-OSTβ to the portal vein. MRP2and MRP3 proteins have a significant role for modified bile acids (glucuronidated or sulphated). Bile acids return to the liver via the portal vein, and are caught at the basolateral membrane by NTCP cholestatic hepatocytes [45]. Another way to limit their intracellular accumulation is by bile acid-induced basolateral hepatocellular export by OSTα-OSTβ, MRP3, or MRP4. In kidneys, adaptive changes in the proximal renal tubules should facilitate renal elimination of bile acids in urine, increasing the total amount of BA along the intrarenal urinary tract. In normal conditions, BA are reabsorbed in the proximal tubule by the ASBT and exported back into systemic circulation, minimizing renal excretion. Under cholestasis, renal clearance becomes the major alternative way for the elimination of divalent sulphated and glucuronidated BA, depending on both increased passive glomerular filtration and active tubular excretion. Transporters for bile acids absorption and excretion are localized in the apical plasma membrane of the proximal renal tubular cells: ASBT for bile acid reabsorption as well as OATP1 and MRP2 for urinary excretion of sulphated/glucuronidated bile salts under cholestatic conditions. Poor data are available about the basolateral transport system, although MRP1 and MRP3 have been localized [42], and there is also little information from human studies about bile acid metabolism and transport in kidneys [46].

Here, we found that BA transport proteins had different localization in cirrhotic rats compared with the control animals. In decompensated cirrhosis, ASBT was not different to healthy rats. On the other hand, in the treated animals, BSEP, MRP-4, and P-Gly partially lost the immunoreactivity highlighted in healthy kidney tubular cells, while NTCP and osteopontin increased it. So, overexposure to BA seems to decrease the immunoreactivity of BSEP, MRP4, and P-Glyc, probably to prevent their systemic increase and to facilitate urinary excretion. This mechanism seems to mimic what happens in the liver, where BAs suppress NTCP to avoid accumulation into hepatitis-infected cells and to reduce their cytotoxic action [47]. NTCP is also decreased in several cholestatic liver diseases [48].

Differently to van Slambrouck et al. [38], we did not find bile casts in distal nephron, nor major pathological lesions suggestive of tubulointerstitial damage.

Moreover, immunohistochemical staining of kidney biopsy specimens was negative for casp-3 in both treated and control rats, suggesting the lack of cell death by caspase-mediated apoptosis. Therefore, the detection of high levels of BA in urine of treated rats may reflect an adaptive renal mechanism in course of cirrhosis.

Previous evidence proved the existence of fine integration between bile acid transporters and inflammatory molecules: while some proinflammatory cytokines can suppress transport systems such as NTCP [49], high bile acids can promote their expression [50].

In treated rats, circulating inflammatory cytokines IL-1, IL-2, IL-4, IL-5, IL-6, IL-7, IL-10, IL-12, IL- 13, IL-17, IL-18, TNF-α, GRO/KC, and RANTES and chemokines G-CSF, M-CSF, MCP-1, and MIP-3α were significantly higher in a phase of compensated cirrhosis (T4) than in decompensated cirrhosis (T12). Urinary biomarkers showed more heterogeneous profiles, although they shared a common tendency towards a zenith after the second week of CCl_4_ inhalation treatment.

Our results showed a nonlinear relationship of cirrhosis evolution with the levels of circulating inflammatory molecules and urinary markers of tubular injury. In particular, the interval between the 4th and 8th weeks following the start of CCl_4_ treatment appears as the critical time in cirrhosis evolution. This concept was also supported by the death of four rats in the following period. On the other hand, we found a linear relation between initial high bile acid levels, feasibly responsible for the tubular injury at T4 and T8, and systemic inflammatory response, followed by a defensive tubular adaptation mechanism. It can be hypothesized that the consequent regression of renal and systemic injury, contributes to bile acid homeostasis, playing important cellular protective role in both kidney and liver. These data are in contrast with previous studies where, even in the absence of bacterial infection, systemic inflammation was correlated with the severity of liver disease and portal hypertension [12].

In a rat model of bile duct ligation (BDL), Shah et al. described an increased expression of TLR4, known as specific activator for the nuclear factor kappa B pathway, which has been implicated in the regulation of multiple biological phenomena including apoptosis. Conversely, our immunohistochemistry analysis showed negative TLR4 expression in both groups, proving a complete recovery of tubular cells [51].

Another BDL experimental study in rodents demonstrated a gradual increase in the serum levels of TNF-α and TGF-β with a respective peak at the 4th and 8th week in cirrhotic rats [52]. Therefore, we can speculate that renal adaptive mechanisms here described are not dependent from the type of liver damage procedure, since the primitive damage is hepatic and the kidney reacts similarly. To the best of our knowledge, this is the first study investigating the profiles of a wide panel of inflammatory molecules in the serum and urinary markers of tubular injury in a rat model of CCl_4_-induced cirrhosis.

The strength of the work is represented by the relevant number of treated animals. Moreover, we carefully documented the cirrhosis development, thus renal findings appear as appropriate and closely representative of what happens in the kidney during hepatic disease. The major limitations of this study are the absence of sacrifices at different time points and the lack of a control through a different model of renal disease, or knockout animals or in vitro experiments. Furthermore, we did not monitor blood pressure over time, nor urinary sodium and creatinine excretion, known to be as relevant findings in hepatorenal syndrome.

## 5. Conclusions

In conclusion, our data seem to suggest the existence of a possible adaptive mechanism, involving BA tubular transporters, through which kidney modulates injury during liver cirrhosis. In our model, the expression of selective transporters appears to be feasibly attenuated by BA-induced kidney damage. Further investigation is needed to provide the applicability of these results in the clinic.

## Figures and Tables

**Figure 1 jcm-11-00636-f001:**
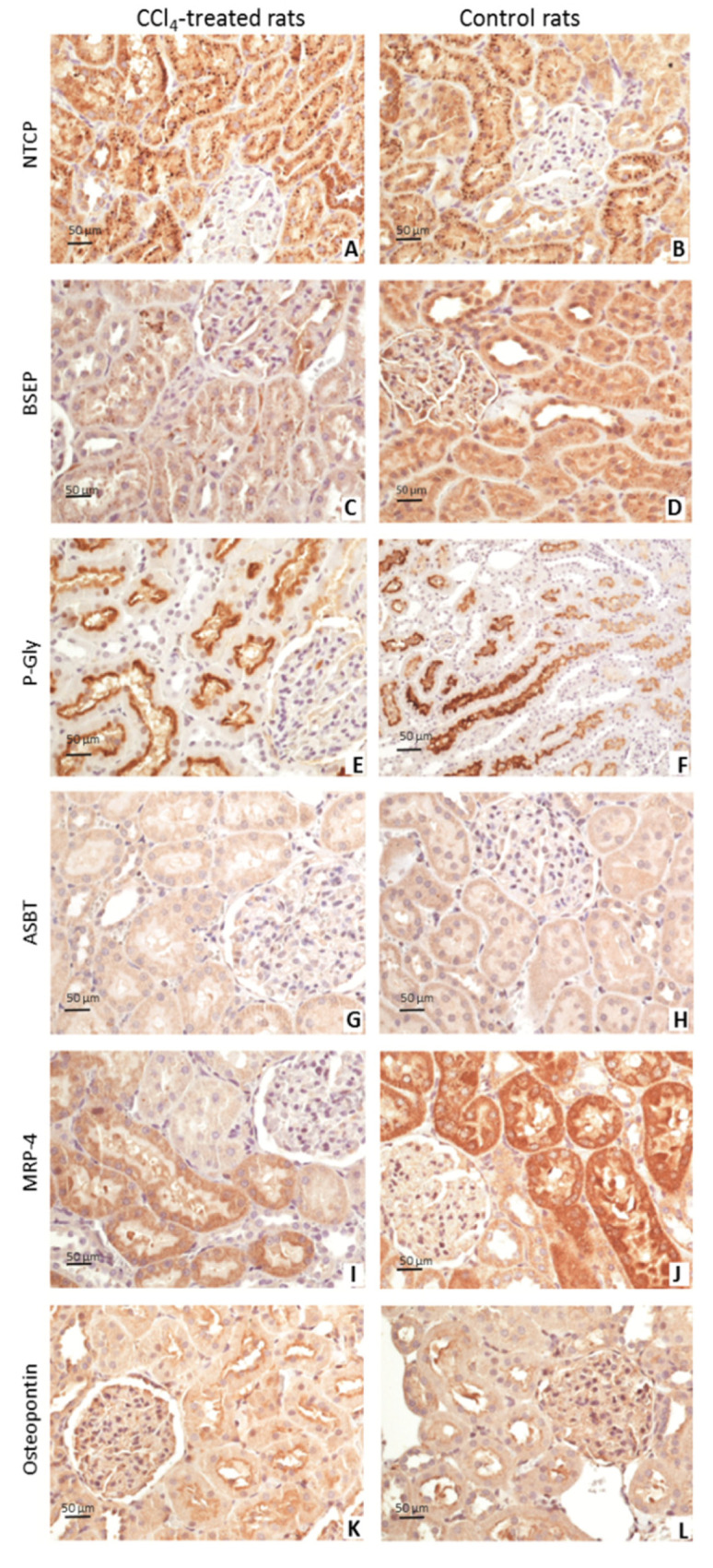
Bile acids transporters in kidneys after 13 weeks of CCl4 treatment. Immunohistochemistry for bile acid transporters in cirrhotic (left) and control rats (right) at 40X (**A**–**C**,**E**,**G**,**H**,**K**,**L**) and at 10× magnification (**F**,**I**,**J**). NTCP showed weak granular cytoplasmic positivity, more intensive in cirrhosis (**A**,**B**). Cytoplasmic positivity for BSEP was weak in cirrhotic and moderate in controls with a granular distribution in both groups (**C**,**D**). P-Gly showed a continuous and thin immunoreactivity always along apical membrane in both groups, more intensive in controls (**E**,**F**). ASBT was weakly positive in both groups (**G**,**H**). MRP4 showed a lower intensity in treated rats than controls, in particular moderate cytoplasm positivity with intense expression on base-lateral membrane (**I**,**J**). For osteopontin, both cirrhotic and control rats showed diffuse apical membrane positivity, more intensive in the treated animals (**K**,**L**).

**Figure 2 jcm-11-00636-f002:**
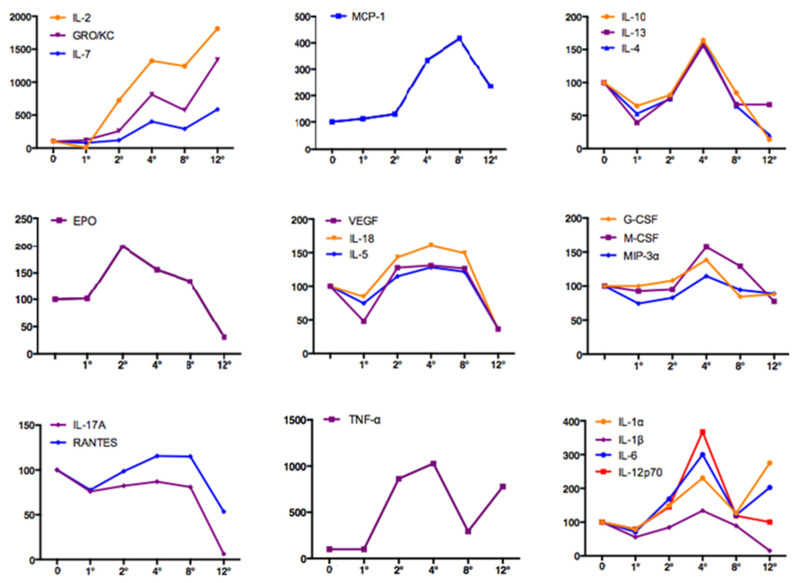
Circulating cytokines and growth factors changes over time during CCl_4_ treatment in the cirrhotic animals. Data are expressed as percentage variation compared with baseline, considering the baseline value as 100%. The comparisons were made between each experimental time vs. T0 by Student *t*-test and *p* < 0.05 was considered statistically significant.

**Figure 3 jcm-11-00636-f003:**
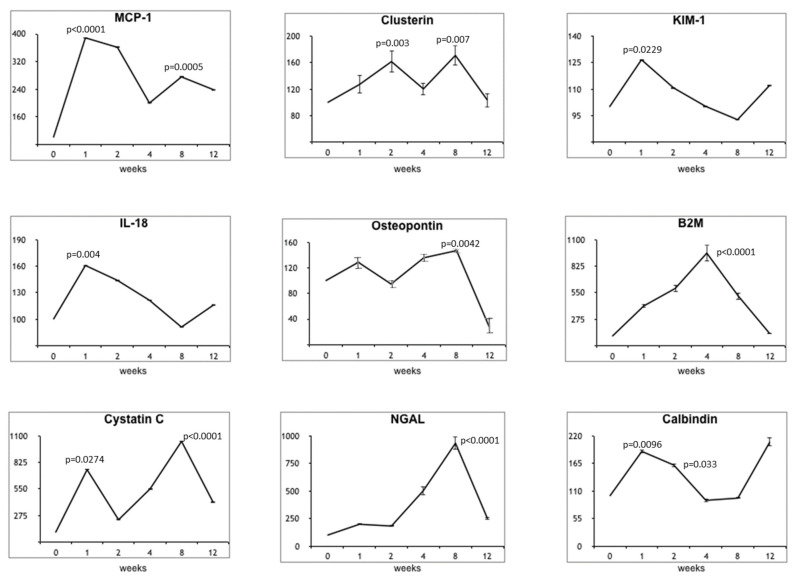
Urinary biomarkers changes over time during CCl_4_ treatment in the cirrhotic animals. Data are expressed as percentage variation compared with baseline, considering the baseline value as 100%. The comparisons were made between each experimental time vs. T0 by Student *t*-test and *p* < 0.05 was considered statistically significant.

**Figure 4 jcm-11-00636-f004:**
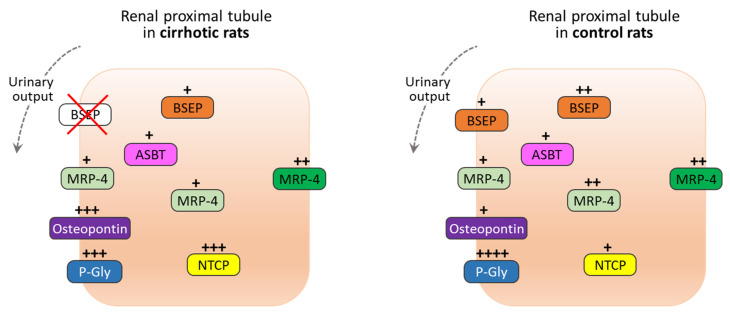
Model of bile acids transport proteins, based on immunohistochemistry findings: transport proteins for bile acids in in jaundice state, such as cirrhosis (**left**) and in healthy renal proximal tubule cell (**right**). The number of “+”on the colored boxes represent the immunohistochemistry staining intensity; +: weak; ++: moderate; +++: intensive; ++++: highly intensive.

**Table 1 jcm-11-00636-t001:** Activity and fibrosis according to Ishak’s scheme [24] and METAVIR system [25] on 21 CCl_4_-treated rats of the cirrhosis model and 5 untreated controls.

CCl_4_-Treated Rats	Total Activity Stage [24]	Fibrosis Stage [24]	METAVIR Stage [25]
1	5	6	F4
2	5	6	F4
3	5	6	F4
4	5	6	F4
5	8	6	F4
6	8	6	F4
7	8	5	F3
8	8	6	F4
9	3	6	F4
10	3	6	F4
11	3	6	F4
12	4	6	F4
13	2	6	F4
14	4	6	F4
15	4	6	F4
16	6	6	F4
17	4	5	F3
18	3	6	F4
19	2	6	F4
20	2	4	F3
21	2	6	F4
**Control rats**			
22	0	0	F0
23	0	0	F0
24	0	0	F0
25	0	0	F0
26	0	0	F0

**Table 2 jcm-11-00636-t002:** Concentration of free, taurine-conjugated, and total bile acids (BAs) in treated and untreated animals. Data are provided as mean values ± SD. The groups were compared by Student *t*-test and a *p* < 0.05 was considered as statistically significant.

	CCl_4_-Treated Rats (*n* = 21)	Control Rats (*n* = 5)	*p*-Value
Free BA in plasma, µmol/L	16 ± 1.5	0.92 ± 0.03	<0.0001
Free BA in urines, µmol/L	81 ± 11.6	0.12 ± 0.06	<0.0001
Taurine-conjugated BA in plasma, µmol/L	10 ± 5	2.1 ± 1.3	0.021
Taurine-conjugated BA in urines, µmol/L	112 ± 20	0.06 ± 0.01	<0.0001
Total BA in plasma, µmol/L	26 ± 17	2.8 ± 1.2	0.0062
Total BA in urines, µmol/L	101 ± 11.2	0.26 ± 0.2	<0.0001

**Table 3 jcm-11-00636-t003:** Summary of transport proteins for bile acids in CCl_4_-treated and control rats detected with immunohistochemistry assay.

		CCl_4_-Treated Rats (*n* = 21)			Control Rats (*n* = 5)		
	Putative Role in Kidney	Cytoplasm	Apical Membrane	BL Membrane	Cytoplasm	Apical Membrane	BL Membrane
NTCP	Unknown	Intensive/granular	/	/	Weak/granular	/	/
BSEP	Unknown	Weak/granular	/	/	Moderate/granular	Weak	/
P-Gly	Unknown	/	Moderate/intensive	/	/	High/intensive	/
ASBT	Reabsorption of free bile acids from lumen to PTC	Weak	/	/	Weak	/	/
MRP4	Transporter for free bile acids from cytoplasm of PTC to lumen	Weak	Weak	Moderate	Moderate	Weak	Intensive
Osteopontin	Transporter for free bile acids from cytoplasm of PTC to serum, through the basolateral membrane	/	Moderate/intensive	/	/	Weak	/

BSEP: bile salt export pump; ASBT: solute carrier family 10 member 2; MRP4: multidrug resistance protein 4; PTC: proximal tubular cells; P-Gly: P-Glycoprotein; NTCP: solute carrier family 10 member 1.

**Table 4 jcm-11-00636-t004:** Total score in decompensated cirrhotic rats during the study. The last row reports the mean ± SD of each score and of the overall score.

CCl_4_-Treated Rats	Histological Score (Activity + Fibrosis)	Biochemical Score	Ascites	Encephalopathy	Overall Score
1	11	7	1	2	21
2	11	13	3	3	30
3	11	6	3	1	21
4	11	0	2	1	14
5	14	10	3	3	30
6	14	6	3	2	25
7	13	9	3	2	27
8	14	7	1	1	23
9	9	5	3	1	18
10	9	7	3	3	22
11	9	6	3	2	20
12	10	7	1	1	19
13	8	5	3	2	18
14	10	5	3	1	19
15	10	7	3	1	21
16	12	6	1	2	21
17	9	5	1	1	16
18	9	7	3	2	21
19	8	8	3	2	21
20	6	6	3	3	18
21	8	4	3	2	17
Mean ± SD	10.3 ± 2.1	6.5 ± 2.4	2.5 ± 0.9	1.8 ± 0.7	21.0 ± 4.1

**Table 5 jcm-11-00636-t005:** Spearman’s rank correlation coefficients and related *p* values for the associations of serum creatinine levels at the various experimental times with inflammatory cytokines and growth factors.

		sCreat T1	sCreat T2	sCreat T4	sCreat T8	sCreat T12
IL-1α	Correlation coefficient	−0.316	0.467	0.409	0.005	0.197
	*p*-value	0.541	0.038 *	0.073	0.982	0.391
IL-6	Correlation coefficient	−0.566	0.424	0.451	0.084	0.181
	*p*-value	0.242	0.063	0.046 *	0.724	0.432
IL-12p70	Correlation coefficient	−0.664	0.359	0.496	0.063	−0.037
	*p*-value	0.150	0.143	0.043 *	0.792	0.873
IL-18	Correlation coefficient	−0.566	0.381	0.466	−0.112	−0.234
	*p*-value	0.242	0.108	0.051 ^	0.638	0.308
GRO/KC	Correlation coefficient	−0.310	0.250	0.468	0.178	−0.033
	*p*-value	0.550	0.288	0.038 *	0.479	0.890
MCP-1	Correlation coefficient	−0.655	0.449	0.360	0.422	0.325
	*p*-value	0.158	0.047 *	0.119	0.064	0.175

sCreat: serum creatinine; IL-1α: interleukin 1 alpha; IL-6: interleukin 6; IL-12p70: interleukin 12p70; IL-18: interleukin 18; GRO/KC: growth-related oncogene/keratinocyte chemoattractant; MCP-1: monocyte chemoattractant protein-1. A *p* < 0.05 was considered statistically significant (* significant, ^ nearly significant).

**Table 6 jcm-11-00636-t006:** Spearman’s rank correlation coefficients and related *p* values for the associations of serum creatinine levels at the various experimental times with urinary biomarkers of kidney injury.

		sCreat T1	sCreat T2	sCreat T4	sCreat T8	sCreat T12
IL-18	Correlation coefficient	−0.100	0.647	0.098	0.194	0.200
	*p*-value	0.873	0.002 *	0.718	0.488	0.475
Osteopontin	Correlation coefficient	−0.600	0.622	0.242	0.089	0.047
	*p*-value	0.285	0.003 *	0.367	0.710	0.879
B2M	Correlation coefficient	0.200	0.328	0.757	0.018	0.041
	*p*-value	0.747	0.171	0.003 *	0.946	0.884
NGAL	Correlation coefficient	0.200	0.108	0.721	0.031	−0.447
	*p*-value	0.747	0.651	0.004 *	0.904	0.095

sCreat: serum creatinine; IL-18: interleukin 18; B2M: beta-2-microglobulin; NGAL: neutrophil gelatinase-associated lipocalin. A *p* < 0.05 was considered statistically significant (* significant).

## Data Availability

Data are contained within the article or Appendix A.

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
