# Peer review of "Adaptive Mechanisms of Renal Bile Acid Transporters in a Rat Model of Carbon Tetrachloride-Induced Liver Cirrhosis"

_jcm, 2022, doi:10.3390/jcm11030636_

Round 1

Reviewer 1 Report

Donadei and Angeletti et al., report a possible mechanism of Acute Kidney Injury developed in an experimental model of decompensated cirrhosis induced by CCL4 inhalation for four weeks. The manuscript is well written, with a correct design and enough experiments to provide interesting information about tubular mechanisms for bile acid transporters in response to cirrhosis-induced AKI. Anyway, I have some major concerns

  • Three CCL4 treated rats are F3, Do the authors think they should be excluded from the analyses?
  • Figure 1 should be clarified; Images are at 10x and 40x magnification, so in line 228 “slides at 40x magnification” should be removed. In the text, it states: “Cytoplasmic positivity for BSEP was weak in cirrhotic and moderate in controls with a granular distribution in both groups (A, B)” but figure A and B correspond to NTCP, so this information needs to be corrected. In the same way, ASBT seems to be G and H.
  • Alterations in NTCP and BSEP expression have been reported in pathophysiological conditions, so a positive and strong expression in the control group would be needed, on the other side, a quantitative assessment of all the bile acids transported would be interesting and would provide more information than those provided by the authors in 3.2 section.
  • Do authors report an increase in creatinine and BIL levels and a decrease in albumin in CCL4-treated rats but it's significant? P values are not provided in tables S4 and S5.
  • Table 2: p values should be provided.
  • Figure 2 and 3: P values should be provided
  • Figure 4: In the text, it states: ”The number of + on the colored boxes represent the immunohistochemistry staining intensity”. How was this intensity quantified? This information is important and is missed in the Materials and methods section.

Author Response

Comments and Suggestions for Authors

Donadei and Angeletti et al., report a possible mechanism of Acute Kidney Injury developed in an experimental model of decompensated cirrhosis induced by CCL4 inhalation for four weeks. The manuscript is well written, with a correct design and enough experiments to provide interesting information about tubular mechanisms for bile acid transporters in response to cirrhosis-induced AKI. Anyway, I have some major concerns

Three CCL4 treated rats are F3, Do the authors think they should be excluded from the analyses?

We thank the Reviewer for the suggestion. We decided to not exclude the 3 rats with METAVIR F3 stage because the overall evaluation of fibrosis was based on both Ishak’s and METAVIR scoring systems. Moreover, even in the presence of milder fibrosis stage, the treated animals really different if compared to the controls, all scoring 0 with the Ishak’s scheme and F0 with METAVIR.

Figure 1 should be clarified; Images are at 10x and 40x magnification, so in line 228 “slides at 40x magnification” should be removed.

This has been done in the amended version of the manuscript.

In the text, it states: “Cytoplasmic positivity for BSEP was weak in cirrhotic and moderate in controls with a granular distribution in both groups (A, B)” but figure A and B correspond to NTCP, so this information needs to be corrected. In the same way, ASBT seems to be G and H.

Many thanks for it. We corrected the mistake.

Alterations in NTCP and BSEP expression have been reported in pathophysiological conditions, so a positive and strong expression in the control group would be needed, on the other side, a quantitative assessment of all the bile acids transported would be interesting and would provide more information than those provided by the authors in 3.2 section.

We have better clarified the differences between treated and untreated rats for all the bile acids transporters in the text section 3.2.

Do authors report an increase in creatinine and BIL levels and a decrease in albumin in CCL4-treated rats but it's significant? P values are not provided in tables S4 and S5.

Tables S4 and S5 report the individual values of serum creatinine and total bilirubin for each single rat in the CCl4-treatement arm, numbered progressively from 1 to 21. The reason why the P values are not provided in these tables is that we could not perform any statistic test to compare the means between T1 and T12 because most of the serum concentrations for both creatinine and bilirubin were undetectable (reported as <0.17 and <0.15 mg/dL respectively, that means below the limit of sensitivity of the assays).

Table 2: p values should be provided.

We added a column with the p values to the table.

Figure 2 and 3: P values should be provided

We have addressed this point (in the text for figure 2 at section 3.4, and inside the plots of each urinary biomarker for figure 3).

Figure 4: In the text, it states: ”The number of + on the colored boxes represent the immunohistochemistry staining intensity”. How was this intensity quantified? This information is important and is missed in the Materials and methods section.

We have corrected the legend of Figure 4 to explicate the meaning of the number of +. The figure indeed mirrors the content of the Table 3, we decided to add these scheme to iconize the overall pattern of bile acid transporters intensity and topographic distribution based on immunohistochemistry.

We performed a semiquantitative evaluation using a specific methodology and the dedicated software ImageJ (NIH), as detailed in the section 2.3 of the amended manuscript. 

Reviewer 2 Report

This paper evaluated kidney and liver histology and circulating and urinary markers of inflammation and tubular injury in a rat model of AKI with liver cirrhosis induction by carbon tetrachloride (CCl4) inhalation. The "story" of this paper itself is understandable; however, the results from this paper are not quite sufficient to prove it. Especially, one “champion” picture of histology was not enough to demonstrate the amount of expression and localization of transporters. Additionally, evaluations of cytokines and biomarkers were of little value without any causal relationships with the phenotypes (AKI and cirrhosis).

Major concerns and questions for authors:

  1. The most significant limitation is that, as described above, the pictures of histology might be “champion” data, and not quite sufficient to demonstrate expression levels and localization of transporters. At least, expression levels should be (semi-)quantified from additional analysis of histology, and by different experimental systems.

  1. Evaluations of cytokines and biomarkers were of little value. Generally, reduced kidney function (or increased SCr) including what seen in AKI lead to increased systemic cytokines and urinary biomarkers. Spearman’s rank Correlations between SCr and biomarkers were almost “natural.” Additionally, we could not know whether these changes were specific to cirrhosis-induced AKI (maybe not). Moreover, time course of those values (e.g. peak in T4 and T8 …) was also specific to “CCI4-induced rat model,” therefore clinical significance was the least.

  1. The authors concluded that “our data provide evidence for a possible adaptive mechanism, involving BA tubular transporters, through which kidney modulates injury during liver cirrhosis. In our model, selective transporters expression attenuates the kidney damage elicited by BA, increasing their excretion through urine and cell metabolism.” However, I do never think that this data clarify the causality between “a possible adaptive mechanism” with “kidney injury.” Other experimental models including in vitro and/or those using knockout models are necessary to demonstrate this causality. Rather, on the contrary, don’t these mechanisms increase bile casts formation and worsen renal function?

Minor concerns and questions for authors:

  1. Time course of blood pressure and urinary Na excretion (with urinary Cr excretion), those decrease are key findings of hepatorenal syndrome, should be clarified.

Author Response

Comments and Suggestions for Authors

This paper evaluated kidney and liver histology and circulating and urinary markers of inflammation and tubular injury in a rat model of AKI with liver cirrhosis induction by carbon tetrachloride (CCl4) inhalation. The "story" of this paper itself is understandable; however, the results from this paper are not quite sufficient to prove it. Especially, one “champion” picture of histology was not enough to demonstrate the amount of expression and localization of transporters. Additionally, evaluations of cytokines and biomarkers were of little value without any causal relationships with the phenotypes (AKI and cirrhosis).

Major concerns and questions for authors:

The most significant limitation is that, as described above, the pictures of histology might be “champion” data, and not quite sufficient to demonstrate expression levels and localization of transporters. At least, expression levels should be (semi-)quantified from additional analysis of histology, and by different experimental systems.

As stated in the section 2.3 of the methods and 3.2 of the results, we performed a qualitative and semi-quantitative assessment of bile acid transporters by immunohistochemistry indeed. In the revised version of the manuscript we have detailed the procedure used for immunohistochemistry scoring based on a specific methodology previously described (reference 23) and the dedicated software ImageJ (NIH), as detailed in the section 2.3. 

Evaluations of cytokines and biomarkers were of little value. Generally, reduced kidney function (or increased SCr) including what seen in AKI lead to increased systemic cytokines and urinary biomarkers. Spearman’s rank Correlations between SCr and biomarkers were almost “natural.” Additionally, we could not know whether these changes were specific to cirrhosis-induced AKI (maybe not). Moreover, time course of those values (e.g. peak in T4 and T8 …) was also specific to “CCI4-induced rat model,” therefore clinical significance was the least.

There is evidence from clinical studies that serum and urinary biomarkers are better and earlier predictors of AKI in cirrhotic patients than serum creatinine, thus in our opinion the correlation is not that obvious. A study by Yap et al found that urinary NGAL and KIM-1 represent useful biomarkers to predict the development of hepatorenal syndrome in patients with advanced cirrhosis whose serum creatinine levels are still within the normal range (Yap DY et al, Dig Liver Dis. 2017; doi:10.1016/j.dld.2016.11.001).

The authors concluded that “our data provide evidence for a possible adaptive mechanism, involving BA tubular transporters, through which kidney modulates injury during liver cirrhosis. In our model, selective transporters expression attenuates the kidney damage elicited by BA, increasing their excretion through urine and cell metabolism.” However, I do never think that this data clarify the causality between “a possible adaptive mechanism” with “kidney injury.” Other experimental models including in vitro and/or those using knockout models are necessary to demonstrate this causality. Rather, on the contrary, don’t these mechanisms increase bile casts formation and worsen renal function?

We completely agree and understand your concern, since in view of the limited number of experimental animals and the lack of a control through in vitro studies or knockout models, we cannot draw any firm conclusion, but rather postulate hypotheses. We have reformulated the conclusions with less assertiveness and broadened the paragraph about the limitations of the study.  

Minor concerns and questions for authors:

Time course of blood pressure and urinary Na excretion (with urinary Cr excretion), those decrease are key findings of hepatorenal syndrome, should be clarified.

We appreciate this is a reasonable concern, but unfortunately blood pressure time course and urinary Na+ and creatinine excretion were not evaluated. Given the importance of such data, we have added this point among the limitations of the study.

Round 2

Reviewer 2 Report

All of the issues that I pointed out were adequately revised. I have no additional comments.